# A Novel Nomogram for Estimating a High-Risk Result in the EndoPredict^®^ Test for Estrogen Receptor-Positive/Human Epidermal Growth Factor Receptor 2 (HER2)-Negative Breast Carcinoma

**DOI:** 10.3390/cancers17020273

**Published:** 2025-01-16

**Authors:** Víctor Macarrón, Itsaso Losantos-García, Alberto Peláez-García, Laura Yébenes, Alberto Berjón, Laura Frías, Covadonga Martí, Pilar Zamora, José Ignacio Sánchez-Méndez, David Hardisson

**Affiliations:** 1Department of Pathology, Hospital Universitario La Paz, 28046 Madrid, Spain; victor.macarron@salud.madrid.org (V.M.); laura.yebenes@salud.madrid.org (L.Y.); alberto.berjon@salud.madrid.org (A.B.); 2Biostatistics Department, Hospital Universitario La Paz, 28046 Madrid, Spain; itsaso.losantos@salud.madrid.org; 3Molecular Pathology and Therapeutic Targets Group, La Paz University Hospital (IdiPAZ), 28046 Madrid, Spain; alberto.pelaez@salud.madrid.org; 4Breast Unit, Department of Gynecology and Obstetrics, Hospital Universitario La Paz, 28046 Madrid, Spain; lfrias@salud.madrid.org (L.F.); covadonga.marti@salud.madrid.org (C.M.); 5Department of Medical Oncology, Hospital Universitario La Paz, 28046 Madrid, Spain; mpilar.zamora@salud.madrid.org; 6Faculty of Medicine, Universidad Autónoma de Madrid, 28029 Madrid, Spain; 7Center for Biomedical Research in the Cancer Network (Centro de Investigación Biomédica en Red de Cáncer, CIBERONC), Instituto de Salud Carlos III, 28029 Madrid, Spain

**Keywords:** breast cancer, EndoPredict, nomogram, prognosis, recurrence

## Abstract

Patients diagnosed with estrogen receptor-positive/human epidermal growth factor receptor 2 (HER2)-negative breast cancer may benefit from adjuvant chemotherapy, subject to a risk assessment for relapse. EndoPredict^®^, a second-generation genomic assay, is widely used to calculate this risk. However, the use of such genomic tests in breast cancer may result in significant economic costs for public healthcare systems. This article presents the development of a predictive nomogram for estimating a high-risk test result based on clinicopathological data from patients who underwent the EndoPredict^®^ test at our center. The objective of this nomogram is to assist clinicians in identifying patients who are most likely to benefit from the test, thereby optimizing its utilization and reducing unnecessary costs.

## 1. Introduction

According to the American Cancer Society, breast cancer is expected to be the most commonly diagnosed neoplasm in the U.S. in 2024, with an estimated 313,510 new cases. Despite this, breast cancer mortality has decreased in recent years due to advances in diagnosis and treatment [1]. Overall, 80% of breast cancers are estrogen receptor (ER)-positive/human epidermal growth factor receptor 2 (HER2)-negative tumors, and a subset of these patients may benefit from adding adjuvant chemotherapy to their treatment regimen, potentially reducing the risk of recurrence by an additional 2–10% [2,3]. However, selection of candidates for adjuvant chemotherapy should be done carefully, considering the risk of recurrence and the potential toxicity associated with chemotherapy [4,5].

The risk of breast cancer recurrence has been traditionally estimated using clinicopathological factors, including age, menopausal status, tumor size, tumor grade, lymph node involvement, and immunohistochemical biomarkers such as the proliferation index (Ki67), estrogen receptor, progesterone receptor, and HER2 amplification [6,7,8]. However, these factors alone do not accurately reflect the biological behavior of ER-positive/HER2-negative breast cancer, which may result in discrepancies between estimated risk of recurrence and actual tumor progression. This may result in over- or undertreatment [9,10]. To address this issue, software tools such as the Nottingham Prognostics Index (https://www.evidencio.com/models/show/633; date of accession: 13 May 2024) or NHS PREDICT (https://breast.predict.cam/tool; date of accession: 13 May 2024) have been developed to stratify the risk of recurrence in these patients using some clinicopathological characteristics [10,11].

Nevertheless, these tools partially elucidate tumor heterogeneity, rendering them inadequate for comprehensive risk stratification in ER-positive/HER2-negative breast carcinoma [12,13]. In recent years, a number of genomic assays have been developed with the aim of more accurately stratifying relapse risk and guiding clinical treatment of these patients. These include assays such as Oncotype DX^®^ (Genomic Health, Inc., Redwood City, CA, USA), MammaPrint^®^ (Agendia NV, Amsterdam, The Netherlands), Prosigna^®^ (Veracyte, South San Francisco, CA, USA), and EndoPredict^®^ (Myriad Genetics, Inc, Salt Lake City, UT, USA). These tests provide independent prognostic information beyond traditional clinicopathological features, offering a more precise estimation of relapse risk in ER-positive/HER2-negative breast carcinoma. This advantage can help avoid overtreatment in patients with a low risk of relapse while ensuring that those who could benefit receive adjuvant chemotherapy treatment [2,6,14]. First-generation genomic assays, such as Oncotype DX^®^ or MammaPrint^®^, are particularly effective for estimating early recurrence risk (0–5 years). However, this has subsequently been identified as a limitation. Second-generation genomic tests, such as Prosigna^®^ and EndoPredict^®^, have demonstrated the capacity to enhance the estimation of late risk (5–10 years) by integrating clinicopathological data into their analyses [6,8].

EndoPredict^®^ is a second-generation genomic test designed to determine the risk of early and late relapses in patients diagnosed with ER-positive/HER2-negative breast carcinoma. This is achieved by analyzing eight genes related to carcinogenesis (BIRC5, UBE2C, DHCR7, RBBP8, IL6ST, AZGP1, MGP, and STC2) and three normalization genes (CALM2, OAZ1, and RPL37A) using a quantitative reverse polymerase chain reaction (RT-PCR) process from formalin-fixed, paraffin-embedded tissue. The evaluation of these genes allows EndoPredict^®^ to provide a molecular index, designated EP. The EPClin index is created by combining the molecular information of the EP index with two clinicopathological variables (tumor size and number of affected lymph nodes). It classifies patients as high or low risk of relapse with greater precision than the EP index for calculating the risk of recurrence in the short (0–5 years) and long term (between 5 and 10 years). The two prognostic indices have been validated in two randomized phase III clinical trials (ABCSG-6 and ABCSG-8), both in pre- and postmenopausal patients [15,16]. Furthermore, the test is decentralized and can be performed in local laboratories in daily clinical practice [17].

The major clinical practice guidelines for breast cancer management, including those from ESMO, ASCO, and St. Gallen, recommend the utilization of EndoPredict^®^ as a prognostic tool for estimating the risk of relapse, alongside other genomic assays such as Oncotype DX^®^, MammaPrint^®^, and Prosigna^®^. However, Oncotype DX^®^ has demonstrated its predictive value in identifying the benefit of adjuvant chemotherapy administration. While Oncotype DX^®^ and MammaPrint^®^ have been clinically validated in prospective randomized trials, Endopredict^®^ and Prosigna^®^ have been retrospectively validated across various studies. Regarding risk estimation, Oncotype DX^®^ and Mammaprint^®^ rely exclusively on genomic tumor profiling, whereas second-generation genomic tests incorporate genomic data and clinicopathological features. In contrast to Oncotype DX^®^, the other three genomic assays offer the advantage of being deployable as local test kits, thereby eliminating the need for centralized laboratory services for result analysis. With respect to test result interpretation, Oncotype DX^®^ necessitates consideration of nodal involvement and patient age, using different cut-off values based on these factors. Conversely, Prosigna^®^ incorporates an intermediate risk category for node-negative patients, which has the potential to complicate adjuvant chemotherapy decision-making [8,12].

Since the implementation of the EndoPredict^®^ test at our center in 2015, we have observed a progressive increase in its use, particularly among patients with an evident high risk of tumor recurrence. Furthermore, the widespread use of genomic assays has the potential to significantly increase healthcare costs in public health systems, as these tools are more expensive than other techniques used in pathology departments [18]. It is therefore essential to develop effective selection criteria for conducting the EndoPredict^®^ test in patients diagnosed with RE-positive/HER2-negative breast carcinoma. Recently, nomograms have been developed for predicting risk categorization in other genomic assays, including Oncotype DX^®^ and MammaPrint^®^. These nomograms are predictive models based on clinicopathological data from patients who underwent the genomic assay and are designed to approximate the risk categorization provided by the assay. These models could assist clinicians in limiting the utilization of genomic assays to patients with an “intermediate risk” of recurrence [19,20,21].

In this study, a novel predictive model is proposed. The model was developed to estimate the probability of obtaining a high-risk result in the EndoPredict^®^ assay. The model is represented by a nomogram, and it was developed based on the clinicopathologic characteristics of patients with ER-positive/HER2-negative breast cancer who underwent the test at our institution. The model was initially developed from a training cohort, in which the relationship between EndoPredict^®^ results and several clinicopathological factors was subjected to multivariate analysis. Subsequently, the predictive model underwent external validation using a separate cohort to assess its predictive power. The primary objective of this nomogram is to facilitate precise patient selection for EndoPredict^®^ testing, thereby enhancing clinicians’ confidence in test requisition and potentially reducing healthcare costs.

## 2. Materials and Methods

### 2.1. Patient and Clinicopathological Feature Selection

The study cohort comprises patients diagnosed with primary ER-positive/HER2-negative breast carcinoma (pT1-3, pN0-1a, M0) who underwent an EndoPredict^®^ genomic test at La Paz University Hospital in Madrid, Spain, between 2015 and 2023. The exclusion criteria were as follows: tumor recurrence, neoadjuvant hormonal therapy/chemotherapy, tumor cryoablation, and male patients. In addition, patients lacking sufficient clinicopathological data were also excluded from the study. The final sample size comprised 348 patients. The clinicopathological features were extracted from the clinical and pathological reports and included age, tumor size, tumor histology, tumor grade, multifocality, presence of ductal carcinoma in situ, presence of lymphovascular space invasion (LVSI), sentinel lymph node (SLN) status, TNM staging, estrogen and progesterone receptor status, Ki67, and HER2 status.

Tumor histology was classified as invasive ductal carcinoma, invasive lobular carcinoma, and other histological subtypes. Tumor grade of differentiation was classified as G1 (well differentiated), G2 (moderately differentiated), and G3 (poorly differentiated). TNM staging was determined in accordance with the 8th edition of the AJCC Cancer Staging Manual [22]. pT stage was grouped into two categories (pT1 and pT2/pT3). The expression of estrogen receptor (clone EP1, Agilent-Dako, Glostrup, Denmark) and progesterone receptor (clone PgR 1294, Agilent-Dako) was determined in accordance with the 2020 ASCO/CAP guideline recommendations [23]. The status of the HER2 oncogene (Herceptest, Agilent-Dako) was evaluated in accordance with the 2023 ASCO/CAP guideline update [24]. The evaluation of Ki67 (clone MIB1, Agilent-Dako) was conducted in accordance with the International Working Group on Ki67 in Breast Cancer recommendations [25].

The EPClin index, as determined by the EndoPredict^®^ test, was calculated by combining the EP genomic index information, the tumor size (pT1ab, pT1c, pT2, or pT3), and the lymph node status (negative, 1–3 affected nodes, 4–10 affected nodes, or more than 10 affected nodes), as previously described [26]. The patients were classified as high-risk or low-risk for tumor relapse, with the cut-off point set at EPClin > 3.3 or EPClin ≤ 3.3, respectively [15].

### 2.2. Statistical Analysis

From the initial cohort of 348 patients, two secondary cohorts were randomly created. The first was a training cohort for predictive model development and internal validation, consisting of 270 patients (approximately 80% of the total). The second was an external validation cohort, consisting of 78 patients (approximately 20% of the total). Age, tumor size, estrogen receptor status, progesterone receptor status, and Ki67 were evaluated as continuous variables. Tumor histology, tumor grade, multifocality, presence of ductal carcinoma in situ, presence of LVSI, SLN status, pT and pN stage, EP risk categorization, and EPClin risk categorization were defined as categorical variables. An initial statistical analysis was performed to determine the existence of significant differences between the two cohorts.

Univariate statistical analysis was performed in the training cohort to determine which clinicopathological features were significantly associated with EPClin risk categorization. Continuous variables were assessed using the Mann–Whitney U test, whereas categorical variables were assessed using the χ^2^ test. To develop a prediction model to estimate the outcome of EPClin, binary logistic regression models were employed. All variables that demonstrated a statistically significant independent association with the outcome of EPClin were considered. The final model was selected using the backward elimination method, and the optimal model was chosen based on the Akaike Information Criterion (AIC) method. The nomogram was used as a visual representation of the final multivariate model. Internal validation of the nomogram was performed using the training cohort, whereas external validation was performed using the external validation cohort and respective receiver operating characteristic (ROC) curves were obtained. A *p*-value of less than 0.05 was considered statistically significant. All statistical analyses were performed using R statistical computing software (version 4.3.3, R Core Team, 2024). The nomogram was estimated using the rms package (version 6.8-0, Frank E, 2024).

## 3. Results

### 3.1. Baseline Clinicopathological Feature Distribution in the Cohort

The baseline clinicopathological features of the 348 patients included in the study are summarized in Table 1. Continuous variables are expressed as medians and interquartile ranges, given that they are not normally distributed. The median age was 53 years (interquartile range, 16). The median size of the tumors was 15 mm (interquartile range, 8), with 59.8% classified as pT1c. The most prevalent histological type was invasive ductal carcinoma (76.1%), followed by invasive lobular carcinoma (15.2%). The majority of tumors were moderately differentiated (63.5%). A multifocal distribution of the tumor was observed in 23.3% of cases. In 78.5% of cases, an accompanying ductal carcinoma in situ was identified. LVSI was detected in 20.6% of the cases. Most cases (63.5%) showed no evidence of lymph node involvement in the SLN study. Following lymph node evaluation, 22.7% of patients were classified as pN1mi stage, and 14.9% as pN1a stage. The median Ki67 was 19% (interquartile range, 15%). The greater part of patients (72.6%) was classified as high-risk for relapse based on the EP molecular index, while the EPClin index classified 55.5% of patients as high-risk for recurrence.

### 3.2. Nomogram Development

Table 2 summarizes the baseline clinicopathological features of the training cohort and the external validation cohort. There were no significant differences in the clinicopathological features between both cohorts.

The univariate statistical analysis performed in the training cohort demonstrated a statistically significant association between the EPClin index and tumor size, pT stage, tumor grade, SLN status, pN stage, and Ki67 (Table 3).

A multivariate statistical analysis was performed to estimate the probability of a high-risk result in the EPClin index, with the objective of generating a predictive model based on the clinicopathological features that demonstrated a statistically significant association with the EPClin index in the univariate statistical analysis (Table 4, Figure 1).

The internal validation of the predictive model was performed using the training cohort of 270 patients (Figure 2a). The area under the curve (AUC) for the training cohort ROC curve is 0.803 (CI95% = 0.751, 0.855), with an optimal sensitivity and specificity at a threshold of 0.536. External validation of the predictive model was carried out using the external validation cohort of 78 patients (Figure 2b). The AUC for the external validation cohort ROC curve was 0.789 (CI95% = 0.689, 0.890), with optimal sensitivity and specificity values observed at a threshold of 0.393.

### 3.3. How to Use the Nomogram

The nomogram is comprised of scoring bars that are associated with clinicopathological variables derived from the predictive model. The initial scoring bar, designated as “Points”, is utilized to allocate individual scores for each clinicopathological variable. The sum of all individual scores from clinicopathological features obtained in the “Points” section is then transferred to the scoring bar labeled “Total Points”. Finally, the probability of obtaining a high-risk result in the EPClin index is determined by extrapolating the total points from the “Total Points” bar to the final bar labeled “Predicted Value”. For example, a patient who obtains a total of 76 points on the “Total Points” bar has a probability > 80% of obtaining a high-risk result on the EPClin index. Predicted values below 0.1 and above 0.9 were not represented on the “Predicted Value” bar.

## 4. Discussion

In this study, we have developed for the first time a novel nomogram for predicting the high-risk EndoPredict^®^ score in a large series of ER-positive/HER2-negative breast carcinomas. This nomogram is based on a predictive model developed utilizing easily obtainable clinicopathological features from pathological reports. The predictive model demonstrates a high degree of accuracy, with a sensitivity and specificity exceeding 70% in the training cohort. This indicates its considerable utility in estimating EPClin results [27]. The clinicopathological variables selected for the nomogram were SLN status, pN stage, tumor size, tumor grade, and Ki67. These features are well-known prognostic markers in ER-positive/HER2-negative breast cancer. Some of these features, such as tumor size, TNM stage, and nodal status, are powerful predictors of early relapse (0–5 years), while tumor grade could play a relevant role in late relapse prediction (5–10 years) [28,29]. Lymph node status has been shown to play an important role in risk prediction by the nomogram; however, it should not be considered in isolation when predicting EPClin results for adjuvant chemotherapy administration decisions. Although patients with lymph node involvement typically derive greater absolute benefit from adjuvant chemotherapy, some with positive nodes may still be classified as low-risk category by the EndoPredict^®^ test [15,16]. Furthermore, Ki67 demonstrates a statistically significant correlation with the EPClin index in our cohort, in a manner analogous to that observed in other published studies [30,31]. This evident correlation is likely attributable to the fact that genomic assays incorporate proliferation genes into their algorithms, reflecting their connection with chemotherapy response in ER-positive/HER2-negative breast carcinoma [16].

The nomogram was developed using a cohort of women diagnosed with ER-positive/HER2-negative breast carcinoma at a tertiary center between 2015 and 2023. The age of the patients ranged from 26 to 79 years, with a median age of 53. Over three-quarters of the tumors were infiltrating ductal carcinoma, and 221 cases (more than 60%) were classified as moderately differentiated (G2). About half of the G2 tumors (113 cases) exhibited a high-risk EPClin result, which aligns with findings from other studies and suggests that using tumor grade alone to predict relapse risk is not reliable [31]. Nearly 80% of the tumors were classified as stage pT1, indicating that most were 2 cm or smaller. SLN analysis showed metastases in 127 cases, while four cases with negative SLNs showed metastases in non-SLNs or in the lymphadenectomy. The EP molecular index identified 95 cases (27.4%) as low-risk, with seven of these subsequently reclassified as high-risk by the EPClin index. On the other hand, 252 cases (72.6%) were initially classified as high-risk by the EP index, but 67 (26.6%) of them were downgraded to the low-risk category in the EPClin index. Overall, 21.3% of the cases were reclassified when clinicopathological features were integrated with the molecular data. This discrepancy between the EP and EPClin indices, which has been observed in other studies, may be attributed to the relevant prognostic impact of tumor size and lymph node status in the EPClin index algorithm [26,32]. This could also explain why Ki67 does not consistently correlate with the EPClin index in some series [31,32,33].

Recent studies designed to evaluate the impact of the EndoPredict^®^ test on clinical decision-making have provided predictive models to estimate EPClin index results. One such model was proposed by Dinh et al. with the objective of assessing changes in chemotherapy recommendations for patients with ER-positive/HER2-negative breast carcinoma based on EndoPredict^®^ results. Their multivariate analysis identified several clinicopathological factors (lymph node metastases, tumor size, Ki67 levels, and progesterone receptor positivity) as significantly associated with EPClin results. Unlike our proposed nomogram, these authors incorporate progesterone receptor levels as a variable to estimate EPClin results [14]. Evaluating progesterone receptor status is crucial for determining breast carcinoma prognosis and helps to categorize these tumors into luminal A or luminal B subtypes, highlighting its relevant role in developing effective prediction models [30]. Another paper published by Almstedt et al. also investigated the impact of EndoPredict^®^ results on clinical decisions and offered a multivariate analysis to estimate EPClin results. This predictive model includes patient age at primary breast surgery, tumor size, nodal status, histological grade, and Ki67 levels as clinicopathological features. Unlike our model, this prediction system considers patient age as an additional prognostic factor for estimating EPClin results, acknowledging that younger age is a relevant negative prognostic factor for breast cancer recurrence and often leads to more aggressive therapy in these patients. However, patient age can sometimes result in overtreatment [34,35]. While both studies provided tools to estimate EPClin results, neither aimed to develop a model specifically for identifying high-risk patients suitable for the EndoPredict^®^ test. Furthermore, neither study provided a nomogram to illustrate their predictive model, which could facilitate its implementation in routine clinical practice.

While there is currently no nomogram designed specifically to estimate EndoPredict^®^ results in patients with ER-positive/HER2-negative breast carcinoma, several nomograms have been published for other genomic assays that are more widely used to assess relapse risk in this disease. Thus, Lee et al. designed a nomogram to predict the likelihood of obtaining a low-risk result in the MammaPrint^®^ test. Using a cohort similar in size to ours, their model demonstrated sensitivity and specificity values similar to those of our predictive model. The clinicopathological features included in their nomogram were age, nuclear grade, progesterone receptor levels, and Ki67 index. Unlike EndoPredict^®^, Mammaprint^®^ relies solely on genomic information to calculate relapse risk in ER-positive/HER2-negative breast cancer, which may explain the limited inclusion of purely clinical factors in the predictive model as occurs in our nomogram [19,36]. Similarly, several nomograms have been developed for the Oncotype DX^®^ test. One of these models, proposed by Lee et al. and using a similar sample size to our cohort, aimed to predict patients who would obtain a low Oncotype DX^®^ recurrence score (ODX RS) by combining clinicopathological variables like estrogen and progesterone receptor levels, nuclear grade, LVSI, and Ki67 index [20]. In another study, Orucevic et al. analyzed data from over 40,000 patients and developed four nomograms to estimate high- and low-risk ODX RS based on both commercial cut-off values and those from the TAILORx trial. These models included six variables, namely age, tumor size, histologic grade, progesterone receptor levels, LVSI, and histologic type [21,37]. A comparison of these two Oncotype DX^®^-based models reveals that, in a larger cohort, clinical variables such as tumor size or age have a greater impact on the prediction of Oncotype DX^®^ results.

The main limitations of our study include its retrospective design and the relatively modest cohort size, as all patients were from a single institution. Increasing the sample size could enhance the accuracy of the predictive model in estimating EPClin results and could help in identifying additional clinicopathological features to be included in the predictive model. Additionally, while our nomogram provides valuable support for clinical decision-making, patient preferences were not addressed. A key strength of this research is the consecutive inclusion of patients, which enabled us to expand the cohort size in a controlled and progressive manner. Furthermore, all patients were managed by the same multidisciplinary team (MDT), ensuring uniformity in treatment protocols and patient selection criteria for the EndoPredict^®^ test. This approach enhances the reliability of the nomogram as a practical tool for clinicians, potentially improving personalized treatment decisions within the MDT framework. The study findings also identify key areas for future research aimed at improving the model’s predictive accuracy and broader applicability.

## 5. Conclusions

In this study, we developed, for the first time, a novel nomogram designed to estimate the probability of obtaining a high-risk result in the EndoPredict^®^ test based on easily accessible clinicopathologic features derived from standard anatomopathologic reports.

The significant increase in EndoPredict^®^ test utilization among patients diagnosed with ER-positive/HER2-negative breast carcinoma highlights the potential clinical utility of this nomogram. It may serve as a valuable resource for identifying patients most likely to benefit from risk stratification for disease recurrence, potentially leading to a reduction in long-term healthcare costs within public healthcare systems.

Despite the modest sample size of our cohort and the retrospective design of this study, the proposed nomogram represents a practical and accessible tool for estimating the probability of obtaining a high-risk EPClin index result in routine clinical practice. In addition, this tool may also increase clinician confidence in ordering the test and reduce patient anxiety regarding treatment decisions, ultimately improving healthcare resource management.

Our research provides a basis for future refinement of this predictive model. Further studies evaluating large patient cohorts are needed to validate the nomogram and elucidate its clinical impact. In addition, the inclusion of additional patient data could potentially improve the quality of the nomogram and facilitate the identification of other clinicopathologic variables that may further improve the accuracy of the EPClin Index outcome prediction.

## Figures and Tables

**Figure 1 cancers-17-00273-f001:**
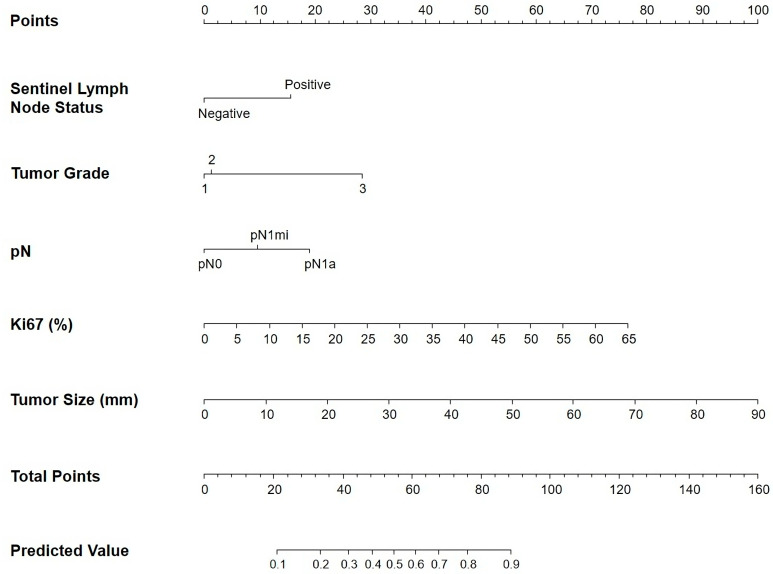
Nomogram to predict for high-risk EndoPredict^®^ score. Sentinel lymph node status, tumor grade, pN stage, Ki67 levels, and tumor size were finally selected to develop the model.

**Figure 2 cancers-17-00273-f002:**
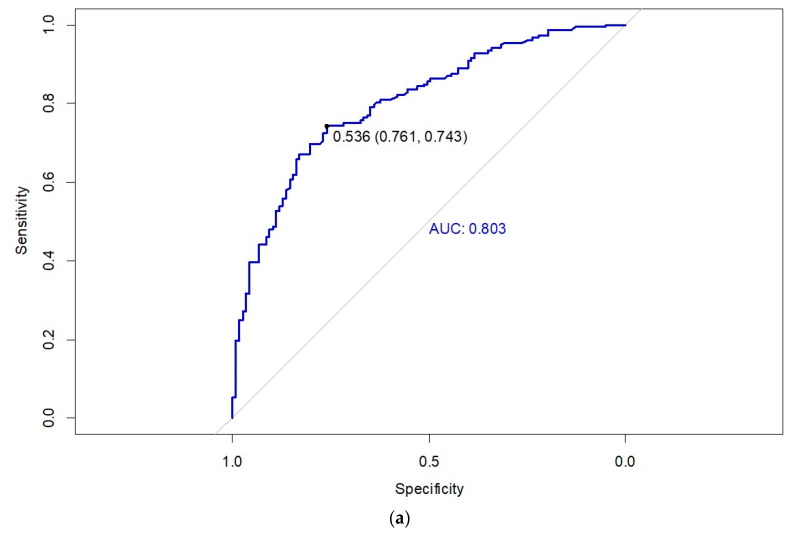
Receiver operating characteristic (ROC) curve of the nomogram for the (**a**) training cohort and (**b**) validation cohort.

**Table 1 cancers-17-00273-t001:** Summary of the clinicopathological features of the study cohort.

Clinicopathological Features	Results
**Age** (years) (n = 348)	53 (IQR, 16)
**Tumor size** (mm) (n = 347)	15 (IQR, 8)
**pT stage** (n = 348)	
pT1a pT1b pT1c pT2 pT3	4 (1.1%) 56 (16.1%) 208 (59.8%) 75 (21.6%) 5 (1.4%)
**Histological tumor type** (n = 348)	
Invasive ductal carcinoma Invasive lobular carcinoma Others	265 (76.1%) 53 (15.2%) 30 (8.6%)
**Tumor grade** (n = 348)	
G1 (well differentiated) G2 (moderately differentiated) G3 (poorly differentiated)	51 (14.7%) 221 (63.5%) 76 (21.8%)
**Multifocality** (n = 344)	
Absent Present	264 (76.7%) 80 (23.3%)
**Ductal carcinoma in situ** (n = 344)	
Absent Present	74 (21.5%) 270 (78.5%)
**LVSI** (n = 344)	
Absent Present	273 (79.4%) 71 (20.6%)
**SLN status** (n = 348) Negative Positive	221 (63.5%) 127 (36.5%)
**pN stage** (n = 348)	
pN0 pN1mi pN1a	217 (62.4%) 79 (22.7%) 52 (14.9%)
**Estrogen receptor** (%) (n = 345)	100 (IQR, 5)
**Progesterone receptor** (%) (n = 344)	75.5 (IQR, 85)
**Proliferation index (Ki67)** (%) (n = 347)	19 (IQR, 15)
**EP score** (n = 347)	
Low-risk High-risk	95 (27.4%) 252 (72.6%)
**EPClin score** (n = 348)	
Low-risk High-risk	155 (44.5%) 193 (55.5%)

IQR, interquartile range. LVSI, lymphovascular space invasion. SLN, sentinel lymph node.

**Table 2 cancers-17-00273-t002:** Baseline clinicopathological characteristics of the training cohort and external validation cohort.

	Results		*p*-Value
	Training Cohort (n = 270)	External Validation Cohort (n = 78)	
**Age** (years) (n = 348)	53 (IQR, 16)	54 (IQR, 14.5)	0.906
**Tumor size** (mm) (n = 347)	15 (IQR, 8)	15 (IQR, 10)	0.719
**pT stage** (n = 348)			
pT1 pT2 + pT3	214 (79.3%) 56 (20.7%)	54 (69.2%) 24 (30.8%)	0.064
**Histological tumor type** (n = 348)			
Invasive ductal carcinoma Invasive lobular carcinoma Others	206 (76.3%) 41 (15.2%) 23 (8.5%)	59 (75.6%) 12 (15.4%) 7 (9.0%)	0.990
**Tumor grade** (n = 348)			
G1 (well differentiated) G2 (moderately differentiated) G3 (poorly differentiated)	40 (14.8%) 173 (64.1%) 57 (21.1%)	11 (14.1%) 48 (61.5%) 19 (24.4%)	0.829
**Multifocality** (n = 344)			
Absent Present	208 (77.9%) 59 (22.1%)	56 (72.7%) 21 (27.3%)	0.344
**Ductal carcinoma in situ** (n = 344)			
Absent Present	55 (20.6%) 212 (79.4%)	19 (24.7%) 58 (75.3%)	0.443
**LVSI** (n = 344)			
Absent Present	213 (79.5%) 55 (20.5%)	60 (78.9%) 16 (21.1%)	0.920
**SLN status** (n = 348)			
Negative Positive	167 (61.9%) 103 (38.1%)	54 (69.2%) 24 (30.8%)	0.223
**pN stage** (n = 348)			
pN0 pN1mi pN1a	164 (60.7%) 60 (22.2%) 46 (17.0%)	53 (67.9%) 19 (24.4%) 6 (7.7%)	0.125
**Estrogen receptor** (%) (n = 345)	100 (IQR, 5)	100 (IQR, 0)	0.437
**Progesterone receptor** (%) (n = 344)	75 (IQR, 85)	80 (IQR, 80)	0.367
**Proliferation index (Ki67)** (%) (n = 347)	20 (IQR, 17)	16 (IQR, 12)	0.090
**EP score** (n = 347)			
Low-risk High-risk	75 (27.8%) 195 (72.2%)	20 (26.0%) 57 (74.0%)	0.754
**EPClin score** (n = 348)			
Low-risk High-risk	118 (43.7%) 152 (56.3%)	37 (47.4%) 41 (52.6%)	0.559

IQR, interquartile range. LVSI, lymphovascular space invasion. SLN, sentinel lymph node.

**Table 3 cancers-17-00273-t003:** Univariate analyses of EPClin stratification of baseline clinicopathological features in the training cohort.

Clinicopathological Features	Results		OR (CI95%)	*p*-Value
	Low-Risk EPClin (n = 118)	High-Risk EPClin (n = 152)		
**Age** (years) (n = 270)	53 (IQR, 14.75)	53 (IQR, 16.25)	0.996 (0.972, 1.020)	0.739
**Tumor size** (mm) (n = 270)	13 (IQR, 8)	17 (IQR, 8)	1.042 (1.012, 1.076)	0.009 *
**pT stage** (n = 270)				
pT1 pT2 + pT3	103 (87.3%) 15 (12.7%)	111 (73.0%) 41 (27.0%)	2.536 (1.350, 4.986)	0.005 *
**Histological tumor type** (n = 270)				
Invasive ductal carcinoma Invasive lobular carcinoma Others	86 (72.9%) 23 (19.5%) 9 (7.6%)	120 (78.9%) 18 (11.8%) 14 (9.2%)	Ref	Ref
			0.561 (0.282, 1.099)	0.094
			1.115 (0.467, 2.787)	0.809
**Tumor grade** (n = 270)				
G1 (well differentiated) G2 (moderately differentiated) G3 (poorly differentiated)	21 (17.8%) 87 (73.7%) 10 (8.5%)	19 (12.5%) 86 (56.6%) 47 (30.9%)	Ref	Ref
			1.093 (0.548, 2.189)	0.801
			5.195 (2.114, 13.536)	<0.001 *
**Multifocality** (n = 267)				
Absent Present	89 (76.7%) 27 (23.3%)	119 (78.8%) 32 (21.2%)	0.886 (0.496, 1.596)	0.684
**Ductal carcinoma in situ** (n = 267)				
Absent Present	23 (19.8%) 93 (80.2%)	32 (21.2%) 119 (78.8%)	0.920 (0.500, 1.671)	0.785
**LVSI** (n = 268)				
Absent Present	98 (83.8%) 19 (16.2%)	115 (76.2%) 36 (23.8%)	1.615 (0.879, 3.042)	0.128
**SLN status** (n = 270)				
Negative Positive	86 (72.9%) 32 (27.1%)	81 (53.3%) 71 (46.7%)	2.356 (1.415, 3.980)	0.001 *
**pN stage** (n = 270)				
pN0 pN1mi pN1a	86 (72.9%) 19 (16.1%) 13 (11.0%)	78 (51.3%) 41 (27.0%) 33 (21.7%)	Ref	Ref
			2.379 (1.289, 4.517)	0.007 *
			2.799 (1.402, 5.871)	0.005 *
**Estrogen receptor** (%) (n = 268)	100 (IQR, 0)	100 (IQR, 7.5)	0.994 (0.974, 1.012)	0.510
**Progesterone receptor** (%) (n = 267)	80 (IQR, 75)	70 (IQR, 88)	0.995 (0.989, 1.001)	0.136
**Proliferation index (Ki67)** (%) (n = 269)	15 (IQR, 17)	22 (IQR, 15)	1.054 (1.029, 1.081)	0.001 *

IQR, interquartile range. LVSI, lymphovascular space invasion. OR, odds ratio. SLN, sentinel lymph node. Ref, reference group for the comparison. *, significant *p*-value.

**Table 4 cancers-17-00273-t004:** Clinicopathological variables included in the multivariate analysis.

	OR (CI95%)	Standard Error	Z Score	*p*-Value
Tumor size	1.068 (1.031, 1.111)	0.19	3.46	<0.001 *
Tumor grade	2.606 (1.517, 4.590)	0.28	3.40	<0.001 *
SLN status	2.880 (0.842, 9.680)	0.62	1.71	0.086
pN stage	1.862 (0.867, 4.251)	0.40	1.55	0.120
Proliferation index (Ki67)	1.078 (1.045, 1.115)	0.02	4.58	<0.001 *

OR, odds ratio. SLN, sentinel lymph node. CI95%, 95% confidence interval. *, significant *p*-value.

## Data Availability

The data presented in this study are not openly available due to confidentiality reasons but are available upon reasonable request from the corresponding authors.

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
