# Peer review of "A Novel Nomogram for Estimating a High-Risk Result in the EndoPredict® Test for Estrogen Receptor-Positive/Human Epidermal Growth Factor Receptor 2 (HER2)-Negative Breast Carcinoma"

_cancers, 2025, doi:10.3390/cancers17020273_

Round 1

Reviewer 1 Report

Comments and Suggestions for Authors

Dear authors, 

the manuscript is fascinating, however, I have some comments and suggestions which could potentially improve the manuscript's quality. 

1. In the introduction section, if possible, please summarize the results of other research papers in one table. After the table please clearly state what are the disadvantages of other research papers (research methodology). After that, you need to clearly indicate the idea and the novelty in this paper. Then you have to clearly indicate the hypotheses and questions that you're going to investigate in this paper. The last paragraph in the intro section should contain a basic description of the structure of your paper.

2. All graphs in paper must contain grids. By adding a grid to a graph you are improving readability of the manuscript. 

3. The conclusion is too short. Generally, the manuscript should consist of 4 paragraphs:

The first paragraph should contain the basic description of what was done in the paper. 

The second paragraph should contain answers to hypotheses defined in the intro section 

The third paragraph should contain advantages and disadvantages of proposed research methodology that was used in this paper. 

The fourth paragraph should contain directions for the future work i.e. to improve the disadvantages of proposed research methodology used in this paper. 

Author Response

Authors’ Response to Reviewers’ Comments

Macarrón et al. A novel nomogram for estimating a high-risk result in the EndoPredict® test for estrogen receptor-positive/human epidermal growth factor receptor 2 (HER2)-negative breast carcinoma (Cancers- 3381695)

Reviewer #1:

Thank you very much for taking the time to review this manuscript. Please find the detailed responses below, and note the highlighted revisions and corrections in the re-submitted files.

Point-by-point response to Comments and Suggestions for Authors

Dear authors, 

the manuscript is fascinating, however, I have some comments and suggestions which could potentially improve the manuscript's quality. 

  1. In the introduction section, if possible, please summarize the results of other research papers in one table. After the table please clearly state what are the disadvantages of other research papers (research methodology). After that, you need to clearly indicate the idea and the novelty in this paper. Then you have to clearly indicate the hypotheses and questions that you're going to investigate in this paper. The last paragraph in the intro section should contain a basic description of the structure of your paper.

According to the reviewer's indications, a more detailed analysis of the remaining genomic tests has been incorporated into the introduction (Please see lines 110-126). Furthermore, we have restructured the final part of the introduction in accordance with the reviewer's recommendations (Please see lines 140-150).

  1. All graphs in paper must contain grids. By adding a grid to a graph you are improving readability of the manuscript.

According to the reviewers' indications, grids have been incorporated into the tables, thereby enhancing their readability.

  1. The conclusion is too short. Generally, the manuscript should consist of 4 paragraphs:

The first paragraph should contain the basic description of what was done in the paper. 

The second paragraph should contain answers to hypotheses defined in the intro section 

The third paragraph should contain advantages and disadvantages of proposed research methodology that was used in this paper. 

The fourth paragraph should contain directions for the future work i.e. to improve the disadvantages of proposed research methodology used in this paper. 

The conclusions have undergone a thorough review and have been restructured in accordance with the reviewers' recommendations (Please see lines 378-398).

Reviewer 2 Report

Comments and Suggestions for Authors

The manuscript, titled "A Novel Nomogram for Estimating a High-Risk Result in the EndoPredict® Test for Estrogen Receptor-Positive/Human Epidermal Growth Factor Receptor 2 (HER2)-Negative Breast Carcinoma", provides an innovative approach to stratifying breast cancer relapse risk using a clinicopathological-based nomogram. By focusing on clinicopathological parameters, the study provides a cost-effective and accessible alternative to genomic assays, enhancing decision-making in resource-limited settings. The authors have meticulously prepared this article. Overall, the paper is well-prepared and written. However, the author should address the following minor issues, which could help improve the significance of the study:

·       The manuscript briefly mentions other genomic tests (e.g., Oncotype DX®, MammaPrint®), but a more detailed comparative analysis would contextualize the nomogram's advantages and limitations.

·       The authors are asked to check the manuscript regarding abbreviations (e.g., "AIC method").

Author Response

 Authors’ Response to Reviewers’ Comments

Macarrón et al. A novel nomogram for estimating a high-risk result in the EndoPredict® test for estrogen receptor-positive/human epidermal growth factor receptor 2 (HER2)-negative breast carcinoma (Cancers- 3381695).

Reviewer #2:

Thank you very much for taking the time to review this manuscript. Please find the detailed responses below, and note the highlighted revisions and corrections in the re-submitted files.

Point-by-point response to Comments and Suggestions for Authors

The manuscript, titled "A Novel Nomogram for Estimating a High-Risk Result in the EndoPredict® Test for Estrogen Receptor-Positive/Human Epidermal Growth Factor Receptor 2 (HER2)-Negative Breast Carcinoma", provides an innovative approach to stratifying breast cancer relapse risk using a clinicopathological-based nomogram. By focusing on clinicopathological parameters, the study provides a cost-effective and accessible alternative to genomic assays, enhancing decision-making in resource-limited settings. The authors have meticulously prepared this article. Overall, the paper is well-prepared and written. However, the author should address the following minor issues, which could help improve the significance of the study:

  • The manuscript briefly mentions other genomic tests (e.g., Oncotype DX®, MammaPrint®), but a more detailed comparative analysis would contextualize the nomogram's advantages and limitations.

According to the reviewer's indications, a more detailed analysis of the remaining genomic tests has been incorporated into the introduction (Please see lines 110-126).

  • The authors are asked to check the manuscript regarding abbreviations (e.g., "AIC method").

Thank you for your suggestion. To facilitate the reading of the manuscript, a section has been included with a list of the abbreviations used in the text just before the references.

Reviewer 3 Report

Comments and Suggestions for Authors

From a biostats and clinical epidemiology point of view, here are some comments for the Authors

- title/abstract, are you dealing with male breast cancer too? please, specify if you treated only FBC or MBC too; even if MBC are excluded at line 131!

- mind that EPClin index is strongly colinear with TNM, so these 2 determinants may not used together; thus, eliminate every crosstab comparing EPClin index versus all its constituents!

- why have you skipped a third cohort, the test one, from your modeling approach?

- continuous covariates have to be reported only as median/IQR, categorical ones as absolute/relative frequencies

- therefore, all inferential analyses have to be redone by a non-parametric approach (i.e. median values inferred by Student T is wrong)

- uni/multi- variate analyses: whch ones!!!??? I suppose binary logistic regression models, am I correct?? state it clearly! In that case, show all the uni/multi- variate ORs in 2 dedicated tables

- which R package has been used for the nomogram?

Author Response

 Authors’ Response to Reviewers’ Comments

Macarrón et al. A novel nomogram for estimating a high-risk result in the EndoPredict® test for estrogen receptor-positive/human epidermal growth factor receptor 2 (HER2)-negative breast carcinoma (Cancers- 3381695).

Reviewer #3:

Thank you very much for taking the time to review this manuscript. Please find the detailed responses below, and note the highlighted revisions and corrections in the re-submitted files.

Point-by-point response to Comments and Suggestions for Authors

From a biostats and clinical epidemiology point of view, here are some comments for the Authors

- title/abstract, are you dealing with male breast cancer too? please, specify if you treated only FBC or MBC too; even if MBC are excluded at line 131!

Given the rarity of male breast cancer compared with female breast cancer, it is generally understood that any reference to breast cancer pertains to female breast cancer. However, to avoid any confusion, as the reviewer rightly points out, it is explicitly stated in the Material and Methods section (please see line 157) that male patients were excluded from this study.

- mind that EPClin index is strongly colinear with TNM, so these 2 determinants may not used together; thus, eliminate every crosstab comparing EPClin index versus all its constituents!

Thank you for your comment. The EPClin index is defined as the dependent variable in this study, not another independent variable. As collinearity is studied between independent variables, we have not considered applying it to this study.

- why have you skipped a third cohort, the test one, from your modeling approach?

The study was carried out with internal and external validation, and a third cohort was not separated, as this was not necessary for the approach that was intended.

- continuous covariates have to be reported only as median/IQR, categorical ones as absolute/relative frequencies

Thank you for your comment. Continuous variables have been represented by median and (p25, p75) interval. Following the recommendations of the reviewer, continuous covariates are now represented as median/IQR; categorical covariates are represented as absolute/relative frequencies.

- therefore, all inferential analyses have to be redone by a non-parametric approach (i.e. median values inferred by Student T is wrong)

We concur with the reviewer's comment and apologize for the unintentional error included in the statistical analysis section. The statistical test employed to assess continuous variables was indeed the non-parametric Mann-Whitney test, as specified in the revised version of the statistical analysis section (Please see lines 193-208).

- uni/multi- variate analyses: whch ones!!!??? I suppose binary logistic regression models, am I correct?? state it clearly! In that case, show all the uni/multi- variate ORs in 2 dedicated tables

Thank you for your comment. The regression model employed is binary logistic regression.

According to the reviewer's indications, the ORs have been included in Tables 3 and 4.

- which R package has been used for the nomogram?

Thank you for your comment. The package utilized was the rms package (version 6.8-0, Frank E, 2024). This information has been incorporated into the text (Please see lines 207-208).

Reviewer 4 Report

Comments and Suggestions for Authors

The article's authors present the development of a predictive nomogram for estimating a high-risk test result based on clinicopathological data from patients who underwent the EndoPredict® test at La Paz University Hospital in Madrid, Spain, between 2015 and 2023 in our center. The results of the work are quite interesting. The text of the work is written in an understandable and precise manner. I checked the statistical part of the work. In my opinion, the statistical part of the work is primitive, but it is done correctly and fairly. I have only three comments on the text.

(1) The title of the article should be shortened somewhat.

(2) The work uses a lot of abbreviations. It would be good, in this case, to supplement the work with a table of abbreviations used.

(3) The reference list of the submitted article should be provided following the MDPI style.

Author Response

 Authors’ Response to Reviewers’ Comments

Macarrón et al. A novel nomogram for estimating a high-risk result in the EndoPredict® test for estrogen receptor-positive/human epidermal growth factor receptor 2 (HER2)-negative breast carcinoma (Cancers- 3381695).

Reviewer #4:

Thank you very much for taking the time to review this manuscript. Please find the detailed responses below, and note the highlighted revisions and corrections in the re-submitted files.

Point-by-point response to Comments and Suggestions for Authors

The article's authors present the development of a predictive nomogram for estimating a high-risk test result based on clinicopathological data from patients who underwent the EndoPredict® test at La Paz University Hospital in Madrid, Spain, between 2015 and 2023 in our center. The results of the work are quite interesting. The text of the work is written in an understandable and precise manner. I checked the statistical part of the work. In my opinion, the statistical part of the work is primitive, but it is done correctly and fairly. I have only three comments on the text.

(1) The title of the article should be shortened somewhat.

We acknowledge the reviewer's concern regarding the length of the article title. However, we believe that shortening it would compromise the integrity of the title, as it would potentially result in the loss of crucial information about the content of the article.

(2) The work uses a lot of abbreviations. It would be good, in this case, to supplement the work with a table of abbreviations used.

Thank you for your suggestion. To facilitate the reading of the manuscript, a section has been included with a list of the abbreviations used in the text just before the references.

(3) The reference list of the submitted article should be provided following the MDPI style.

We would like to express our sincere apologies for this unintentional error in the format of the references and would like to thank the reviewer for his/her comment. The reference list has been revised and formatted according to the MDPI style.

Round 2

Reviewer 3 Report

Comments and Suggestions for Authors

The Authors were able to solve most of previous concerns